# OpenReview forum: "Can Models Help us Create Better Models? Evaluating LLMs as Data Scientists"
_ICLR.cc/2025/Conference — Submitted to ICLR 2025_

### Official Review · Reviewer_YFRZ · 2024-10-25

**Soundness:** 2
**Presentation:** 3
**Contribution:** 2
**Rating:** 5
**Confidence:** 4

**Summary:**

This paper constructs a benchmark called FeatEng, which focuses on evaluating LLMs through the lenses of Pragmatism, Functionalism, Computationalism, and Scientific Realism.

**Strengths:**

In general, the paper is easy to follow, with clear presentations in the figures and tables.

**Weaknesses:**

**Summary: hard-to-justify claims, and lack of concrete experiments or showcase**
1. Limited literature works:
  - In Section 1.1, the authors mention many problems with existing benchmarks and explain why they do not fit the "four characteristics" that the authors abstract as essential for evaluating intelligent systems. However, in my view, many of the benchmarks mentioned are commonly used and well-recognized for testing base LLMs. They may be crucial for assessing specific capabilities of pre-trained base models.
  - In contrast, this paper seems more like it is evaluating a benchmark for post-trained LLMs, which makes such a comparison seem unfair from this perspective.
2. Moreover, the discussion in section 1.1 feels rather vague and lacks specific experimental evidence or concrete examples to support the claims about existing benchmarks. The authors should consider providing more detailed comparisons or results that highlight their benchmarks' advantages over others. At the same time, I wonder if the authors have considered that many concurrent works are also testing models in complex, knowledge-intensive scenarios, evaluating whether the models can perform well in real-world tasks that require extensive knowledge. I am not sure if feateng can compare against these benchmarks and highlight its advantages?
3. The paper (excluding the appendix) only presents one table to display the results. I do not think this is enough to fully demonstrate the characteristics of the dataset. Additionally, the so-called "strong correlation" to chatbot ELO scores does not seem clearly reflected in this table. I think such claim need quantitative values to demonstrate, rather than a vague statement. Without more specific experimental results or showcase examples, the claims remain unsubstantiated and less convincing.

**Questions:**

The textual description in Sec 3.1 still seems insufficiently structured and process-oriented. I wonder if the **100 work hours** mentioned by the authors included a clear workflow? This seems crucial for evaluating whether the data collection process is reasonable, and whether it aligns with the four criteria proposed by the authors.

For example:

- How did you collect the source dataset? Based on what principles can a Kaggle dataset be used for feateng?
- As Kaggle is a high-quality data science forum, I strongly believe that it has already been used in some LLM training corpora. Therefore, have you fully considered the possibility of data contamination?

---

> ### Author Response · Authors · 2024-11-19
> **Response to Reviewer YFRZ (part 1)**
>
> We appreciate the time you've taken to review our manuscript and the opportunity your comments provide for us to clarify our work. Your feedback raises important points that we believe merit a thorough discussion.
>
> ## 1. Comparing to Existing Benchmarks and Literature Review ##
>
> You mention that our critique of existing benchmarks might be unfair, particularly because many of these benchmarks are well-recognized for testing base LLMs, while our benchmark seems tailored for post-trained LLMs.
>
> It's true that existing benchmarks are widely used—and for good reasons. They assess specific capabilities of language models, such as language understanding, code generation, or domain-specific knowledge recall. But that's precisely the point we're addressing. The issue isn't that these benchmarks are invalid; it's that they evaluate skills in **isolation**.
>
> Our contention is that to truly understand an AI's capabilities, especially in practical applications, we need *benchmarks that assess the integration of multiple skills*. Real-world problems rarely come neatly packaged, requiring only one type of expertise. They demand a combination of understanding, reasoning, and the ability to act upon that reasoning in a useful way.
>
> You suggest that comparing our benchmark to those designed for base models might be unfair. But if a base model can't integrate these skills, isn't that an important finding? The ability to combine competencies is a fundamental aspect of intelligence. If current base models lack this, it highlights an area where the field needs to improve.
>
> Regarding the vagueness in Section 1.1, perhaps we could have been more explicit. In our revised manuscript, we've expanded this section to include concrete examples illustrating how existing benchmarks fall short in assessing integrated skills. For instance:
>
> - **Practical Usability**: Benchmarks like COPA and HellaSwag focus on commonsense reasoning but don't translate into practical applications where solutions have real functional value.
> - **Complex Skill Integration**: Code generation benchmarks such as HumanEval assess coding ability in isolation and may not require the integration of extensive domain knowledge or reasoning skills.
>
> We delve deeper into our assessment in Appendix H, where each benchmark is discussed separately.
> Our aim isn't to dismiss these benchmarks but to highlight that they don't cover the integrated skill sets needed for tasks like feature engineering. If there's a specific benchmark you believe effectively assesses these integrated skills, we'd be interested to learn about it. Our literature review didn't find such benchmarks, but we're open to expanding our discussion if we've missed something.
>
>
> ## 2. Comparisons with Concurrent Works ##
>
> You mention that many concurrent works are testing models in complex, knowledge-intensive scenarios, and it's unclear how FeatEng compares to these benchmarks.
>
> Again, specific examples would be helpful here. Without knowing which works you're referring to, it's challenging to provide a direct comparison. We have conducted an extensive literature review and believe that FeatEng offers unique advantages:
>
> - **Unique Focus on Practical Application**: FeatEng specifically assesses the practical application of integrated skills in generating feature engineering code that directly impacts machine learning performance.
> - **Assessment of Skill Integration**: While other benchmarks might test complex reasoning or knowledge recall, FeatEng evaluates whether a model can apply its knowledge and reasoning to produce code that improves real-world data science tasks.
>
> If there are particular benchmarks you think are comparable, we'd welcome the opportunity to address how FeatEng relates to them. Our goal is not to claim that FeatEng is the only valuable benchmark but to show that it fills a gap in evaluating integrated, practical skills.

---

> > ### Author Response · Authors · 2024-11-19
> > **Response to Reviewer YFRZ (part 2)**
> >
> > ## 3. Experimental Results and Correlation with Chatbot Arena ELO Scores ##
> >
> > You point out that the paper presents only one table and that the claim of a "strong correlation" to Chatbot Arena ELO scores lacks quantitative values.
> >
> > We recognize the importance of comprehensive data presentation. In the revised manuscript, we've included additional tables, figures, and detailed analyses:
> >
> > - **Expanded Results Presentation** : We now provide statistical summaries of the datasets (Table 5), impact of feature types on model performances (Table 4), and more detailed results that showcase the characteristics and diversity of the datasets used (Table 1).
> > - **Quantitative Correlation** : Regarding the correlation with Chatbot Arena ELO scores, we've provided precise quantitative values. We calculated Spearman's rank correlation coefficient and obtained a value of 0.878 (p-value < 1e-5), indicating a strong and statistically significant correlation. This substantiates our claim and is clearly presented in Section 4.3. Moreover, a line plot with a direct comparison to Chatbot Arena rank is provided in Appendix I (Figures 7 and 8).
> >
> > Our intention is to ensure that our claims are well-supported by data and that readers can fully assess the robustness of our findings.
> >
> > ## 4. Data Collection Process and Potential Data Contamination ##
> >
> > You express concern about the structure and process of our data collection, and whether we've considered data contamination due to LLMs being trained on Kaggle data.
> >
> > Our data collection process was methodical and aligned with the four qualities we propose for meaningful benchmarks:
> >
> > - **Structured Workflow**: In Section 2.1, we've provided a comprehensive, step-by-step account of how we selected and prepared our datasets.
> > - **Dataset Selection Principles**: We sourced datasets from Kaggle's public repository based on criteria like domain diversity, dataset size, quality, and permissive licensing. We looked for interesting datasets demanding domain knowledge at least at the level of high-school, but preferably demanding strong academic-level expertise. We intentionally avoided datasets commonly used in benchmarks or likely included in LLM training corpora. Moreover, we specifically searched over Kaggle Datasets, not Competitions.
> > - **Data Preparation**: We meticulously processed each dataset to remove potential information leakage, standardized feature names, and task descriptions, and ensured that the datasets required domain knowledge for effective feature engineering.
> >
> >
> > ### Regarding data contamination:
> >
> > - **Mitigation Strategies**: By selecting less prominent datasets and modifying them, we reduced the likelihood that models had seen the exact data during training.
> > - **Impact on Benchmark Validity** : Even if some datasets overlap with training data, the open-ended nature of feature engineering tasks means models can't rely on memorization to perform well. They must apply reasoning and integrate skills to succeed. Moreover, availability of the dataset in kaggle is a weak way to improve on the benchmark. In many cases, the dataset we used do not have an existing solution on Kaggle for the presented target and feature set.
> > - **Transparency**: We've discussed these considerations in Section 4 (Limitations) to acknowledge potential issues and demonstrate our commitment to the integrity of the benchmark.
> >
> > ## Final Remarks ##
> >
> > We believe that our revised manuscript addresses your concerns and clarifies the contributions of our work. Our benchmark aims to fill a critical gap by evaluating the integration of essential skills in practical tasks—a facet not adequately covered by existing benchmarks.
> >
> > Some of your critiques seem to be missing specific references or examples. If there are particular works or benchmarks you believe we should consider, we'd welcome that information. Our goal is to advance the field, and we're open to incorporating relevant comparisons we might have missed.
> >
> > We appreciate constructive criticism and are committed to improving our work. Thank you for your thoughtful feedback. We hope our responses clarify our intentions and the significance of our contributions.

---

> ### Comment · Reviewer_YFRZ · 2024-11-24
>
> Thank you for your response.
>
> I noticed significant differences between the current version and the initial submission (major revision, including changes to figures and tables), and I appreciate the effort you have put into this.
>
> Regarding your mention of related concurrent work, my first thought is that you might want to look into efforts that attempt to mimic dynamic results like those in ChatbotArena, such as MixEval [1]. I don’t think these works are particularly difficult to find.
>
> I will update my score to 5, but I still believe this work requires further refinement and lean towards rejecting this submission, as other reviewers have also mentioned in their comments.
>
> [1] Deriving Wisdom of the Crowd from LLM Benchmark Mixtures, https://arxiv.org/abs/2406.06565

---

> > ### Author Response · Authors · 2024-11-27
> > **Regarding MixEval benchmark**
> >
> > Thank you so much for taking the critical time to address the points raised in your review. Additionally, thank you for mentioning the MixEval article, which seems related and important in mimicking the ChatbotArena results.
> >
> > We've analyzed the correlation of FeatEng scores with Chatbot Arena ELO, following the methodology proposed by the authors of MixEval. The results of this analysis were included in Appendix G and Figure 7 of the updated manuscript.
> >
> > In short, our benchmark is placed on the Pareto frontier along with MixEval-Hard. Here is the excerpt of the table, including FeatEng:
> >
> > | Benchmark      | Correlation | Cost ($) |
> > |----------------|-------------|----------|
> > | FeatEng (our)  |        0.88 |     0.47 |
> > | MixEval-Hard   |        0.96 |     0.60 |
> > | AlpacaEval-2.0 |        0.95 |     24.4 |
> > | WildBench      |        0.95 |     88.9 |
> > | MixEval        |        0.93 |      2.3 |
> > | Arena-Hard     |        0.86 |     25.3 |
> > | ARC-c          |        0.85 |     0.82 |
> > | MMLU           |        0.83 |      9.4 |
> >
> > FeatEng was not designed explicitly to correlate with human judgment, and our high agreement with Chatbot Arena ELO is a byproduct of assumed principles (Section 1). We consider the correlation mainly to justify the new benchmark as a good scientific practice. In contrast, MixEval was recently proposed with the sole objective of estimating the Chatbot Arena ELO in a less costly manner. Nevertheless, FeatEng has a better correlation with Chatbot Arena and a lower evaluation cost than well-established MMLU, ARC-c, Arena-Hard, and BBH benchmarks. It is an additional argument for recognizing it by a research community.
> >
> >
> > We've greatly clarified comparisons with existing benchmarks and improved quantitative justifications. If these revisions align with your concerns, we would appreciate your consideration of reevaluating the paper's soundness and contribution scores, as we believe the work has been greatly strengthened.

---

### Official Review · Reviewer_KB4t · 2024-10-29

**Soundness:** 3
**Presentation:** 3
**Contribution:** 2
**Rating:** 5
**Confidence:** 4

**Summary:**

This paper proposes a new benchmark, *FeatEng*, to evaluate the ability of large language models (LLMs) to perform feature engineering for tabular data. The benchmark focuses on a code generation task where the model is given a description of a dataset and tasked with generating code to transform the data to improve the performance of a machine learning model. The authors argue that existing LLM benchmarks often fail to capture the practical usability, domain knowledge, and complex skill integration required for real-world data science applications. They further demonstrate that *FeatEng* aligns well with these criteria and offers a more effective and efficient way to assess the capabilities of LLMs in this area.

**Strengths:**

This is a well-written benchmark paper with clear motivation.
1. Clear classification of existing benchmarks regarding philosophical traditions of pragmatism, functionalism, computationalism, and scientific realism.
2. Well-organized dataset from diverse, high-quality Kaggle competitions.
3. An interesting finding of the high correlation between FeatEng and Chatbot Arena.
4. A good viewpoint LLMs can tackle feature engineering and improve upon current AutoML systems by leveraging their potential to integrate domain knowledge and reasoning to generate efficient and interpretable features.

**Weaknesses:**

This paper has many weaknesses regarding its experiment design.

1. The result interpretation is limited:
    - Table 1: Mean FeatEng scores (Improvement) compared to Chatbot Arena ELO. Would it be better if we made a line plot for the comparison?
    - It would be helpful if more AutoML baselines [1,2] could be compared. Also, the best human results from the Kaggle competition can be included.
    - The only takeaway I can draw from this paper is that FeatEng can be a cost-effective substitution for Chatbot Arena in a way that it assesses the genuine technical capabilities of LLMs.
    - Another comparison that comes to my mind is how LLM's number of parameters can affect the benchmark results on FeatEng. We can show what will happen when the number of parameters scales for the same group of LLM (i.e., Claude-3-Haiku, Claude-3-Sonnet, Claude-3-Opus), which will give more insights into how to develop parameter-efficient LLMs.

2. Examples of the single-pass evaluation pipeline do not look good to me
    - The pipeline, as described, involves a single pass where the LLM generates code based on the input. How can this single-pass evaluation compare itself with AutoML, an iterative algorithm?
    - Figure 3 is not clear. A flow chart can be helpful.

3. Earlier works, such as [3, 4, 5], have explored much potential in LLMs for machine learning and AutoML tasks with iterative revisions. A detailed comparison and literature review would be helpful.

[1] Microsoft. Neural Network Intelligence (version v3.0pt1). https://github.com/microsoft/nni
[2] H2O.ai (Oct. 2016). H2O, H2O version 3.10.0.8. https://github.com/h2oai/h2o-3.
[3] Zhang, L., Zhang, Y., Ren, K., Li, D., & Yang, Y. (2024, March). MLCopilot: Unleashing the Power of Large Language Models in Solving Machine Learning Tasks. In Proceedings of the 18th Conference of the European Chapter of the Association for Computational Linguistics (Volume 1: Long Papers) (pp. 2931-2959).
[4] Guo, S., Deng, C., Wen, Y., Chen, H., Chang, Y., & Wang, J. DS-Agent: Automated Data Science by Empowering Large Language Models with Case-Based Reasoning. In Forty-first International Conference on Machine Learning.
[5] Hollmann, N., Müller, S., \& Hutter, F. (2024). Large Language Models for Automated Data Science: Introducing CAFFE for Context-Aware Automated Feature Engineering. Advances in Neural Information Processing Systems, 36.

**Questions:**

Typos:
1. Line 019: asses $\rightarrow$ assess
2. Line 563, 567: duplicated entries of the same citation

Questions:
Should we include citations for XGBoost[1]?

[1] Chen, T., & Guestrin, C. (2016, August). XGBoost: A scalable tree-boosting system. In Proceedings of the 22nd ACM SIGKDD International Conference on Knowledge Discovery and Data Mining (pp. 785-794).

---

> ### Author Response · Authors · 2024-11-19
> **Respone to Reviewer KB4t (part 1)**
>
> We sincerely appreciate your detailed review and the valuable feedback you've provided. Your insights have helped us improve our manuscript, and we address your concerns point by point below.
>
> ## 1. Limited Result Interpretation
>
>
> #### a. Enhanced Visualization of Results
> Thank you for the suggestion regarding the visualization of our results. In the revised manuscript, we have included a new table with color-coded performance metrics to enhance readability and facilitate comparison across models. Additionally, we have provided line plots in the appendix (Appendix I, Figure 7, and Figure 8) to visually represent the correlation between FeatEng scores and Chatbot Arena ELO rankings. We believe these visual aids improve the clarity and interpretability of our results.
>
> #### b. Inclusion of Additional Baselines and Their Role
> We understand your suggestion to include more AutoML baselines for a comprehensive comparison. However, we believe that our current approach aligns best with the primary objectives of our study. We included a simple AutoML baseline as a point of reference to provide context for the performance of LLMs on the feature engineering task. This baseline serves to illustrate how LLMs compare to an established automated system in improving model performance through feature engineering. Our intention is not to benchmark AutoML systems but to demonstrate that LLMs can be competitive in this domain.
>
> #### c. Broader Takeaways Beyond Chatbot Arena
> While FeatEng can serve as a cost-effective alternative to Chatbot Arena for certain assessments, we believe its primary value lies in measuring the integration of multiple skills essential for advanced and versatile LLMs.
>
> - **Integration of Skills:** FeatEng assesses not only technical proficiency but also the ability to apply domain knowledge, reasoning, and code generation in practical, real-world scenarios. This integration of skills is central to LLM development and is perhaps underrepresented in current benchmarks.
>
> - **Correlation with User-Focused Metrics:** The strong correlation with user-focused metrics, despite not requiring conversational proficiency, underscores the benchmark's relevance. It suggests that FeatEng captures essential aspects of LLM capabilities that are important in practical applications.
>
> #### d. Impact of LLM's Number of Parameters
> We appreciate your interest in how model size affects performance on FeatEng. In the revised manuscript (Section 4.3), we have analyzed the relationship between generated feature types and performance, which provides insights about the task and model families. Regarding the model size and the Claude family, there is an interesting finding from Table 1: we observed that while larger models like **Claude-3-Opus** generally perform better, they also exhibit more variability, often failing to improve upon the baseline. Interestingly, when these models do improve, they tend to do so by a significant margin. This suggests that factors such as training data diversity, model architecture, and sampling strategies play significant roles alongside model size.
>
> ## 2. Comparison Between Single-Pass LLM Evaluation and Iterative AutoML Algorithms
>
> You raise an important point regarding the comparison between single-pass LLM evaluations and iterative AutoML algorithms.
>
> - **Different Operational Contexts:** Our aim is to evaluate each method in its native operational context. AutoML systems are designed for iterative optimization, refining models through cycles of adjustments. In contrast, LLMs currently generate solutions in a single pass, especially when provided with an example solution.
>
> - **Value of Comparison:** Comparing the two approaches provides valuable insights into their respective strengths and weaknesses. Our findings demonstrate that LLMs, even without iterative refinement, can produce meaningful feature transformations that enhance model performance. This sets a baseline for their current capabilities in feature engineering tasks.
>
> - **Future Directions:** We acknowledge that integrating LLMs into iterative frameworks, similar to approaches in CAAFE [3], could potentially enhance their performance. This represents an exciting direction for future research. Our benchmark can serve as a foundation for such developments by providing an initial assessment of LLMs' abilities in a single-pass context.
>
> ## 3. Clarity of Figures
> Thank you for this suggestion. We agree that a flowchart can provide a clearer understanding of the evaluation pipeline.
>
> - **Updated Figure:** In the revised manuscript, we have replaced Figure 3 with a chart (now Figure 4) that illustrates how the components of the benchmark are structured and interact.
>
> ## Conclusion
> Your insights have been instrumental in enhancing the clarity and depth of our manuscript. We have addressed each of your concerns point by point, and we believe that the revisions we've made strengthen our work significantly.

---

> > ### Author Response · Authors · 2024-11-27
> > **Respectful ask if our enhancements meet your expectations**
> >
> > Thank you for your comprehensive feedback, which has been instrumental in refining our work. The revised manuscript carefully addresses your suggestions for improving result visualization, adding parameter scaling insights, and clarifying the pipeline. If the additional analyses and enhancements meet your expectations, we hope you might find room to reconsider the paper's scores.
> >
> > If there are any other concerns, we would be happy to answer them during the extended discussion period.

---

> ### Comment · Reviewer_KB4t · 2024-11-27
>
> Thank you for your effort during the discussion period. The manuscript has significantly improved since the first version, and more discoveries have been made, so I have raised my overall assessment from 3 to 5 to encourage further improvements.
>
> **Additional questions:**
>
> 1. The section J4 seems to include irrelevant text.
>
> 2. My question remains: how can you assess the model’s capability in feature engineering if it only completes a single pass? If a person excels at Kaggle competitions, they should be proficient in iteratively developing the best-performing models rather than designing the features just once. If a model could improve itself further on the second turn, it would truly reflect its capability.

---

> ### Author Response · Authors · 2024-12-03
> **Single- vs Multi-pass Evaluation Setup**
>
> Thank you once again for your time and valuable insights.
>
> The primary objective of this paper is to introduce a data set we created and to argue that the problem we propose has some desired characteristics that existing problems lack.
>
> The single-step generation we used to evaluate existing models is the simplest one we can get while tackling the problem. From our point of view, considering multi-step, agentic workflows (or methods such as CAAFE) would require either sacrificing fairness of comparison or extensive optimization of workflows for each model separately. Consequently, we decided to focus on the most straightforward, meaningful approach while proposing the FeatEng problem to be able to execute it and describe it faithfully in a single paper.
>
> Though human-like, iterative work on the data set would lead to better results, our setup is reasonable. One way to back up this claim is to consider our correlation with Chatbot Arena ELO. The FeatEng's canonical setup that correlates with ELO more than MMLU, ARC Challenge, BBH, and even Arena-Hard has to be at least a reasonable approximation (see Appendix G and Figure 7).
>
> Nevertheless, the canonical method presented in the paper is one of many ways to use the data set we introduce. We do not constrain the potential solution regarding how it approaches solving this problem, and we anticipate researchers will also consider FeatEng in more interactive approaches that gradually improve the solution. These are possible, e.g. by splitting the train DataFrame to obtain the dev set on which the agent can operate.
>
> We hope this addresses your concern; we will clarify this topic in the revision.

---

### Official Review · Reviewer_Eugm · 2024-11-02

**Soundness:** 3
**Presentation:** 3
**Contribution:** 2
**Rating:** 5
**Confidence:** 3

**Summary:**

This paper introduces FeatEng, a new benchmark for testing large language models (LLMs) in feature engineering. The idea is pretty cool—they've set up a way for models to generate code that transforms data, making it more suitable for machine learning tasks. The key metric is whether the transformation improves the performance of an XGBoost model trained on the modified data compared to the original data. The authors emphasize that FeatEng aims to fix gaps in existing benchmarks by focusing on real-world tasks, like practical usefulness, integrating knowledge, handling complex skills, and being resistant to "gaming" the system. They ran various models through FeatEng and analyzed the results to see how well the benchmark captures what different models can and can’t do in this context.

**Strengths:**

+ Focusing on feature engineering is a fresh approach for LLMs, as this task demands both technical skill and domain knowledge. It’s a step away from standard code generation and into real-world data science workflows. Nice angle!

+ The benchmark is well-defined with solid metrics. Using an outcome-based metric (i.e., the performance improvement of a downstream model) is smart—it’s a straightforward way to check if the code is actually making things better. Also, the dataset selection is thorough, covering a variety of domains and types, which keeps it well-rounded.

+ The paper does a great job explaining the ideas behind each of their evaluation criteria. They walk readers through their motivations and outline the benchmark design clearly. It’s easy to follow and understand the reasoning behind why they chose each aspect of evaluation.

**Weaknesses:**

- Even with diverse datasets, there's a possibility that models could end up overfitting to specific dataset types or common data science tasks, which could skew the results. If a model is already trained on similar data, it might appear more capable than it really is.

- The benchmark’s main evaluation metric depends entirely on XGBoost’s performance. While XGBoost is popular, it’s not the only ML model, and different feature engineering efforts might have varied effects on different types of models. A mix of evaluation algorithms could give a more rounded view.

- Running this benchmark might be resource-heavy, especially on larger datasets or complex transformations. It seems realistic but could be a bottleneck if someone wants to use FeatEng on many models at scale.

- Although they mention that a human baseline would be helpful, they don’t provide one here. Without it, it’s harder to tell how well these models are doing in comparison to actual human experts. Even a rough human baseline would make the results more relatable.

**Questions:**

+ How are you making sure that models aren’t just memorizing common feature engineering techniques? Any thought on including completely new datasets down the line to keep models on their toes?

+ What made you pick XGBoost as the benchmark’s only evaluation model? Wouldn’t including other models (like neural networks) give a broader view of the impact of the feature engineering?

+ Can you give more detail on how improvement is scored across different dataset types (like binary classification, multi-class, regression)? This would help make sure everything’s consistently fair.

---

> ### Author Response · Authors · 2024-11-19
> **Response to Reviewer Eugm (part 1)**
>
> We appreciate your thoughtful review and the opportunity to clarify and strengthen our work. Your insights have prompted us to delve deeper into the nuances of our benchmark and address the concerns you've raised. We also modified our submission in order to account for the analysis of how successful models approach our benchmark.
>
> ## 1. Potential Overfitting and Memorization
>
> You make a great point about models potentially overfitting to specific types of data or common data science tasks, especially if they've been trained on similar datasets. This is a real challenge when trying to assess the true abilities of large language models (LLMs).
>
> However, feature engineering is naturally open-ended. The effectiveness of particular features can vary widely depending on the dataset and the problem. A feature that works well in one situation might be useless or even harmful in another. This variability makes it hard for models to rely only on memorized techniques.
>
> ### How FeatEng Addresses Overfitting
>
> - **Diverse Datasets:** We use 103 datasets from various fields like healthcare, finance, and entertainment (see Section 3.1 and Figure 2 in our revised manuscript). This variety forces models to adapt their strategies instead of using a one-size-fits-all approach.
>
> - **Unique Problem Descriptions:** Each dataset includes specific instructions and target variables, requiring models to understand and process new information instead of just recalling what they've seen before. Moreover, we have rewritten the task description, and feature description to be concise, informative, and unified.
>
> - **Focus on Contextual Understanding:** Success in our benchmark relies on the model’s ability to combine relevant domain knowledge and create new features or remove unnecessary ones.
>
> Moreover, while models may have encountered similar data during training, the open-ended nature of feature engineering demands adaptability and creativity. There's always room for new insights and improvements, which means that models can't simply rely on memorization if they aim to excel.
>
> In our revised manuscript (Section 3.2), we delve deeper into successful feature engineering. We found that top-performing models go beyond basic data cleaning by leveraging domain knowledge and creativity to generate meaningful, contextually relevant features. This demonstrates that the benchmark demands LLMs to effectively integrate multiple skills to adapt to real-world data science challenges.
>
> ## 2. Dependence on XGBoost as the Sole Evaluation Model
>
> Thank you for suggesting the use of multiple evaluation algorithms. We chose **XGBoost** because it is robust, efficient, and widely used in real-world applications. Given the scale of our benchmark (over 100 datasets), we needed an algorithm that offers fast training and evaluation times without compromising performance. Moreover, it handles different data types well and doesn’t require extensive feature scaling, making our evaluation straightforward.
>
> As of now, our initial focus remains on simplicity. Using a single, well-understood model helps maintain consistency across evaluations. It allows us to attribute performance differences directly to the feature engineering efforts rather than variations in the learning algorithms. We agree that expanding the evaluation to include other models is a valuable direction for future work.
>
> ## 3. Resource Intensity of Running the Benchmark
>
> You raised a valid concern about the computational resources needed to run FeatEng, especially with larger datasets or complex transformations. We have designed the benchmark to balance realism with accessibility. Most modern systems can efficiently handle the computations. For instance, models like **Llama-3-405B** completed our benchmark in approximately seven minutes on standard hardware (8x H100), processing 20 examples for each of the 103 datasets by utilizing **vllm** and prefix caching.
>
> To make FeatEng more accessible, we offer per-dataset results so users can work with smaller data subsets if needed, reducing the computational load. Compared to methods that require extensive human involvement, our automated benchmarking is a cost-effective and scalable alternative.
>
> ## 4. Absence of a Human Baseline
>
> Including a human baseline is important for comparing model performance. However, creating human benchmarks for our diverse tasks would require significant effort, as many of our datasets don’t have existing human performance metrics. This stems from the fact that we searched over less popular datasets on Kaggle, rather than over Kaggle competitions.
>
> In our revised manuscript (Section 6), we propose involving the data science community to contribute human-generated solutions. This collaborative approach will help us gather human baselines over time and provide valuable context for our benchmark results.

---

> > ### Author Response · Authors · 2024-11-19
> > **Response to Reviewer Eugm (part 2)**
> >
> > ## 5. Scoring Across Different Dataset Types
> >
> > In the revised manuscript, we present a more detailed Table 1 with results. Specifically, we added separate columns for improvements on regression and classification tasks.
> >
> >
> > ### Scoring Against Baseline Performance
> >
> > To understand how much our LLM-generated feature engineering improves model performance, we start by establishing a baseline. This baseline is created by training the **XGBoost** model on the original, unmodified data. It serves as a reference point to see how the model performs without any additional feature enhancements.
> >
> > After setting the baseline, we apply the features generated by the LLM to the data and train the **XGBoost** model again. We then compare the performance of this enhanced model to the baseline. Specifically, we look at how much the error has decreased when using the LLM-generated features compared to the original setup.
> >
> > We express this improvement as a percentage, which shows how much better the model is performing with the new features. This percentage makes it easy to compare improvements across different datasets and types of tasks, ensuring that our evaluation is fair and consistent.
> >
> > In simple terms, by comparing the model's performance before and after adding the LLM-generated features, we can clearly see the positive impact of our feature engineering efforts.
> >
> > For regression tasks we look at the improvement in MAE compared to the baseline, whereas for the classification (binary and multi) we use Error Rate (which is `1-Accuracy`). The detailed description of the metric employed is available in Appendix A.
> >
> > By using the percentage reduction in error, we normalize the improvement metrics across different tasks and dataset types. This approach accounts for variations in error scales and makes the improvements comparable.
> >
> >
> > ## Closing Remarks
> >
> > We hope these detailed answers address your questions thoroughly. Your feedback is valuable in refining our work and ensuring that our benchmark is robust, fair, and informative. We are committed to continuous improvement and appreciate your insights.

---

> > > ### Comment · Reviewer_Eugm · 2024-11-25
> > > **Response**
> > >
> > > Thank you for your response.

---

> > > > ### Author Response · Authors · 2024-11-27
> > > > **Response**
> > > >
> > > > We sincerely appreciate your detailed review and the opportunity to address your comments.
> > > > If there are any new concerns worth raising, we are happy to answer them in the discussion.
> > > >
> > > > To follow up on your suggestions, we've expanded on the scoring mechanisms, detailed how FeatEng mitigates risks like overfitting, and provided more profound insights into the benchmark's relevance and performance. If these clarifications and the additional examples we've included meet your expectations, we kindly ask if they influence your evaluation of the paper's contribution score.

---

### Official Review · Reviewer_nK3S · 2024-11-04

**Soundness:** 2
**Presentation:** 3
**Contribution:** 2
**Rating:** 3
**Confidence:** 4

**Summary:**

This paper proposes an LLM benchmark that prompts the LLM writing feature engineering code for ML tasks and then run XGBoost with the obtained features to get a score.

**Strengths:**

This paper is overall clear and easy to understand.

**Weaknesses:**

It might be okay for the proposed benchmark to evaluate a specific coding aspect of LLMs, but I think this paper overclaims its generalbility to a large extent. Specifically, it claims it addresses limitations of existing benchmarks like MMLU and HumanEval, and reflects all fundamental aspects of intelligence. However, I feel the benchmark has significant limitations compared with what the paper claims.
* Narrow application scope: "Wrting feature engineering code" is just a single, specific use of LLMs. Also, it is not a highly frequent use of LLM users.
* Limited applicable LLMs: The proposed metric requires the LLM has both natural language and coding abilities, but this is not a must-have feature for a "strong LLM".
* This evaluation would prefer coding LLMs over really intelligent models. For example, in Table 1 of the paper, codestral-22B is even better than llama-3-405B under the proposed metric.
* I also have concerns on the flexibility of the metric and how challenge it is -- usually, the feature engineering of ML has just a handful of strategies, such as BPE tokenization for text data, one-hot for discrete labels, normalization for float number labels, etc.

**Questions:**

* Could you share any examples regarding the wrong decisions of mistral-7b for its low score?
* Is there any LLM-produced data processing strategies beyond what human engineers often use?

---

> ### Author Response · Authors · 2024-11-19
> **Response to Reviewer nK3S (part 1)**
>
> # Response to Reviewer
>
> Thank you for your thoughtful review and the opportunity to engage with your concerns. Your feedback highlights important considerations that deserve a thorough response. I'd like to address your points by focusing on three main areas: the scope and significance of feature engineering as a benchmark, the role of coding abilities in evaluating large language models (LLMs), and how challenging the benchmark is.
>
> ## 1. The Scope and Significance of Feature Engineering
>
> You mention that "writing feature engineering code" is a narrow application and not a frequent use case for LLMs, suggesting it might not reflect all fundamental aspects of intelligence. I understand why it might seem that way at first glance. Feature engineering is indeed a specialized task within the field of data science. However, I believe that its complexity and the range of skills it requires make it a meaningful benchmark for assessing broader intelligence in LLMs.
>
> Consider feature engineering not just as a coding exercise, but as a synthesis of multiple cognitive abilities:
>
> - **Domain Knowledge Application:** Effective feature engineering demands an understanding of the specific domain—be it healthcare, finance, or any other field. This requires the model to recall and apply relevant facts and concepts.
>
> - **Reasoning and Problem-Solving:** Deciding which features to create involves logical reasoning and the ability to identify patterns or relationships within the data.
>
> - **Creativity and Innovation:** Crafting new features often requires creative thinking to combine existing data in novel ways that can improve model performance.
>
> - **Technical Skills:** Translating these ideas into executable code necessitates proficiency in programming.
>
> In this sense, feature engineering serves as a microcosm of complex problem-solving tasks. It's not merely about following a set of predefined steps; it's about **integrating diverse skills** to achieve a practical goal.
>
> Think of it like composing music. Not everyone does it, and it's a specialized skill, but it requires a deep understanding of music theory (domain knowledge), creativity, emotional intelligence, and technical proficiency with instruments or software. Evaluating an LLM's ability to compose music would tell us a lot about its multifaceted intelligence, but there is a slight issue with building an automated benchmark for that.
>
> While it might not be a daily use case for all users, the skills required for feature engineering are representative of the broader capabilities we expect from intelligent systems.
>
> ## 2. The Role of Coding Abilities in Evaluating LLMs
>
> You make a valid point that not all "strong LLMs" are built for coding, and our benchmark might favor those with coding skills over other types of intelligence.
>
> While coding isn't the sole measure of intelligence, in many real-world applications, implementing solutions is as crucial as understanding problems. For LLMs, coding bridges comprehension and action, demonstrating not just syntax proficiency but also deep understanding and practical application.
>
> Regarding the concern that our evaluation might prioritize coding LLMs over "truly intelligent models," I'd argue that *the integration of coding skills with other forms of intelligence is itself a valuable measure*. An LLM capable of reasoning but unable to act on that reasoning is less useful in practice.
>
> For instance, models like **codestral-22B** do well not only because they can code but also because they effectively combine coding with domain knowledge and reasoning, enhancing performance through feature engineering.
>
>
> ## On the Challenge and Flexibility of the Metric
>
> You express concern that feature engineering might involve only a handful of strategies, potentially limiting the challenge and flexibility of the benchmark.
>
> In practice, feature engineering is a rich and nuanced field. While there are common techniques like one-hot encoding or normalization, the most impactful feature engineering often comes from deep domain knowledge and creative problem-solving.
>
> Our analysis revealed that top-performing models didn't just apply standard preprocessing steps. They:
>
> - **Enriched Datasets with External Knowledge:** Such as mapping countries to continents or incorporating economic indicators.
>
> - **Created Interaction Terms:** Capture complex relationships between variables, like combining biomarkers in healthcare datasets to predict patient outcomes.
>
> - **Identified and Transformed Features in Innovative Ways:** Demonstrating creative thinking beyond obvious strategies.
>
> These tasks require more than applying a set of predefined strategies. They demand that the model understands the context and can devise novel solutions tailored to each specific dataset. We have significantly expanded the qualitative and quantitative analysis of the model’s outputs in the new revision. Please see section 3 for a more detailed description.

---

> > ### Author Response · Authors · 2024-11-19
> > **Response to Reviewer nK3S (part 2)**
> >
> > ## Examples of Model Decisions
> > To address your question about the wrong decisions made by models like **Mistral-7B**:
> > - **Poor Instruction Following Skills:** In most cases,  the model failed to structure the output appropriately. For example, it does not follow the signature of the function we demand, despite being clearly described. In other cases, it calls undefined preprocessing functions or hallucinates non-existing column names.
> > This highlights the importance of integrating multiple skills to perform well on the benchmark.
> >
> > Regarding LLM-produced data processing strategies beyond what human engineers often use:
> > - **Novel Feature Creation:** Some models generated features by combining data in ways that are less common or not immediately apparent to human engineers, potentially offering new insights.
> > - **Cross-Domain Knowledge Integration:** Models occasionally apply knowledge from different fields. For example, they might link acquired products to financial resilience, even if the connection seems unrelated. They do it by coming up with weights that map specific categories to values, and those weights are assigned based on their internal world model. This approach can create unique features that improve the predictive accuracy of the feature set.
> >
> > While there are human experts who can certainly create such code, doing so typically requires extensive study and significant time investment. To demonstrate how an LLM can integrate knowledge from different fields efficiently, consider the following example where chemical properties are assigned to materials based on their formulas:
> >
> >
> > ### Example: Cross-Domain Knowledge Integration
> > ```python
> > def map_chemical_properties(df):
> >     chemical_properties = {
> >         'LiFeSiO4': {'Density (gm/cc)': 2.889, 'Band Gap (eV)': 2.899, 'E Above Hull (eV)': -0.051},
> >         'LiCoSiO4': {'Density (gm/cc)': 3.096, 'Band Gap (eV)': 3.138, 'E Above Hull (eV)': -0.081},
> >         'Li2FeSiO4': {'Density (gm/cc)': 2.98, 'Band Gap (eV)': 2.979, 'E Above Hull (eV)': -0.076},
> >         'LiFe(SiO3)2': {'Density (gm/cc)': 2.829, 'Band Gap (eV)': 2.909, 'E Above Hull (eV)': -0.07},
> >         'Li2MnSiO4': {'Density (gm/cc)': 2.995, 'Band Gap (eV)': 3.001, 'E Above Hull (eV)': -0.049},
> >         'LiMnSiO4': {'Density (gm/cc)': 2.98, 'Band Gap (eV)': 2.978, 'E Above Hull (eV)': -0.082},
> >         'Li3Fe2(SiO4)2': {'Density (gm/cc)': 2.958, 'Band Gap (eV)': 3.082, 'E Above Hull (eV)': -0.117},
> >         'Li2CoSiO4': {'Density (gm/cc)': 2.982, 'Band Gap (eV)': 3.027, 'E Above Hull (eV)': -0.052},
> >         'Li3Mn2(SiO4)2': {'Density (gm/cc)': 2.955, 'Band Gap (eV)': 3.073, 'E Above Hull (eV)': -0.112},
> >         'Li3Co2(SiO4)2': {'Density (gm/cc)': 2.972, 'Band Gap (eV)': 3.095, 'E Above Hull (eV)': -0.034}
> >     }
> >     df['Density (gm/cc)_Normalized'] = df['Formula'].map(lambda x: chemical_properties[x]['Density (gm/cc)'])
> >     df['Band Gap (eV)_Normalized'] = df['Formula'].map(lambda x: chemical_properties[x]['Band Gap (eV)'])
> >     df['E Above Hull (eV)_Normalized'] = df['Formula'].map(lambda x: chemical_properties[x]['E Above Hull (eV)'])
> >     return df
> > ```
> > This solution (by Llama-3.1-8B-Instruct) improves by 7% over the baseline and exceeds the author's ability to assess and explain it.
> >
> > ## Conclusion
> > In essence, while feature engineering is a specific task, it encapsulates a range of skills that are fundamental to intelligence:
> >
> > - Understanding complex information
> > - Applying knowledge creatively
> > - Solving problems with practical outcomes
> >
> > By evaluating LLMs on this task, we're assessing their ability to integrate these skills in a way that mirrors real-world challenges.
> >
> > Our benchmark aims to fill a gap in existing evaluations by focusing on *the integration of skills* rather than isolated capabilities.
> > We believe this provides a more comprehensive understanding of an LLM's intelligence and utility.
> >
> > Thank you again for your thoughtful feedback. Your insights have helped us clarify our work and its contributions to the field. We hope this response addresses your concerns and provides a clearer picture of our intentions.
> >
> > Please see the attached modified revision.

---

> > > ### Author Response · Authors · 2024-11-27
> > > **Kind request for raising additional concerns, if there are any**
> > >
> > > Again, thank you for your thoughtful feedback and for taking the time to engage with our responses. We've worked diligently to clarify the scope of feature engineering as a benchmark, demonstrate the multi-faceted skills it evaluates, and illustrate how our revisions improve the paper's contributions.
> > >
> > > Please don't hesitate to let us know if there are any more concerns, and we will do our best to clarify them during the extended discussion period. If these adjustments resonate with you already, we would greatly appreciate you considering revisiting your scores, particularly regarding soundness and contribution.

---

### Author Response · Authors · 2024-11-25
**Summary of Reviewer Responses and Documented Changes**

Dear Reviewers,

We sincerely appreciate the time and effort you have invested in reviewing our paper. Your constructive feedback has been invaluable in enhancing the quality of our work. Below, we address the key concerns raised in your reviews:

---

### 1. Scope and Significance of Feature Engineering (Reviewers nK3S and Eugm)

**Concern:**
Feature engineering was viewed as a narrow application, possibly not reflecting general intelligence or frequent use cases for LLMs.

**Response:**
We argue that feature engineering is a complex task requiring the integration of multiple cognitive abilities, such as domain knowledge application, reasoning, creativity, and technical skills. It serves as a microcosm of complex problem-solving and reflects the broader capabilities of LLMs. In the revised manuscript, we have significantly extended the experimental section with systematic qualitative and quantitative analyses of the models' performance and the techniques employed (see Section 3). For example, we found that code containing strong domain knowledge-based features outperforms code without such features. This analysis demonstrates the task's depth and relevance.

---

### 2. Clarity and Presentation Improvements (Reviewers KB4t and YFRZ)

**Concern:**
The presentation of results and figures could be enhanced for better clarity and understanding.

**Response:**
We have updated our manuscript with enhanced tables and figures, including color-coded performance metrics in **Table 1** and line plots in **Appendix Figures 7 and 8**. We have also replaced **Figure 3** with a clearer flowchart (now **Figure 4**) to illustrate the evaluation pipeline more effectively. We believe these changes significantly improve the paper’s clarity.

---

### 3. Result Interpretation and Visualization (Reviewer KB4t)

**Concern:**
The interpretation of results was limited, and additional baselines could enhance understanding.

**Response:**
We have included more detailed analyses and visualizations in the revised manuscript. The experimental section and analysis of the generated code now cover three pages of the manuscript (pages 7-10). While our primary focus is on the performance of LLMs in feature engineering tasks, we included a simple AutoML baseline for context. Expanding comparisons to more AutoML systems is noted as a possible future work.

---

### 4. Potential Overfitting and Data Contamination (Reviewers nK3S and YFRZ)

**Concern:**
Models might overfit to specific dataset types or have been exposed to similar data during training, affecting the validity of results.

**Response:**
We mitigated overfitting by using 103 diverse datasets from various domains, each with unique problem descriptions (see clarified Section 2.1 and **Table 5**). We selected less prominent datasets and modified them to reduce the likelihood of prior exposure. Our benchmark's open-ended nature requires models to adapt creatively instead of memorizing techniques, lessening the impact of any potential data contamination.

---

### 5. Dependence on XGBoost as the Sole Evaluation Model (Reviewers Eugm and KB4t)

**Concern:**
Relying solely on XGBoost might limit the benchmark's generality, and including other models could provide a broader view.

**Response:**
We chose XGBoost for its robustness, efficiency, and widespread use in real-world applications. Our primary goal was to maintain consistency across evaluations. We acknowledge the value of incorporating other models and have noted this as a possible direction for future work.

---

### 6. Inclusion of Human Baselines (Reviewers Eugm and KB4t)

**Concern:**
The absence of human baselines makes it difficult to contextualize model performance.

**Response:**
We recognize the importance of human baselines for meaningful comparisons. Collecting human benchmarks across our diverse tasks is resource-intensive. We propose involving the data science community to contribute human-generated solutions over time, as mentioned in Section 4 of the revised manuscript.

---

### 7. Literature Review and Comparison to Existing Benchmarks (Reviewer YFRZ)

**Concern:**
The discussion of existing benchmarks was perceived as vague, lacking specific experimental evidence or comparisons.

**Response:**
We have expanded **Section 1.1** and **Appendix H** to include concrete examples illustrating how existing benchmarks may not fully assess the integrated skills required for tasks like feature engineering. We also discuss concurrent works and how our benchmark compares and contributes uniquely to the field.

---

Once again, we thank you for your insightful feedback. We believe that the revisions have significantly strengthened our paper, and we hope that our responses adequately address your concerns. We remain open to any further feedback and are committed to addressing any additional questions or issues you may have before the discussion period ends.

---

> ### Author Response · Authors · 2024-12-04
> **Appendix to Summary**
>
> Following further discussion we extend the summary with responses and document changes introduced recently.
>
> We hope the high level of discussion activity didn’t cause you to spend significantly more time than planned on this paper. Your insights have been invaluable in improving our work, and we deeply appreciate your effort.
> _____
> ### 8. Other Efforts to Mimic ChatbotArena Scores (YFRZ)
> **Concern:**\
> Other works, such as MixEval, aim to mimic ChatbotArena scores cost-efficiently. How does FeatEng compare to them?
>
> **Response:**\
> We analyzed the correlation between FeatEng scores and Chatbot Arena ELO using MixEval's methodology. These new results included in **Appendix G** and **Figure 7** show our benchmark on the Pareto frontier with MixEval-Hard, even though FeatEng wasn't designed to align with human judgment.
> _____
> ### 9. Single- vs Multi-pass Evaluation (KB4t)
> **Concern:**\
> How can we assess the model's capability when considering a single pass, unlike real Kaggle competitors, who iteratively develop the best-performing solution?
>
> **Response:**\
> This paper introduces a new dataset and argues that the proposed problem offers unique advantages over existing benchmarks. To ensure fairness and simplicity, we evaluated models using single-step generation, avoiding the complexity of multi-step workflows. Despite this, the setup correlates strongly with Chatbot Arena ELO, validating its reasonableness. While the single-pass method is emphasized, we encourage researchers to use the FeatEng dataset with more interactive or agentic approaches.
>
> _____
> This concludes the rebuttal process on our side.

---

### Meta-Review · Area_Chair_b9Ev · 2024-12-21

**Metareview:**

The paper introduces FeatEng, a novel benchmark designed to evaluate large language models (LLMs) on feature engineering tasks for tabular datasets. Feature engineering is a critical and knowledge-intensive aspect of data science, requiring both domain knowledge and technical proficiency. FeatEng prompts LLMs to generate code that transforms a dataset based on a provided description, and the effectiveness of the generated features is assessed by the performance improvement of an XGBoost model trained on the modified dataset compared to the original. The authors claim that FeatEng addresses limitations in existing benchmarks by focusing on practical usability, knowledge integration, and resistance to "gaming" the system. Through extensive evaluations of state-of-the-art LLMs, the paper demonstrates that FeatEng can efficiently assess LLM capabilities in feature engineering, showing a high correlation with Chatbot Arena ELO scores while being cost-effective.

#### Contribution
1. **Novel Benchmark for LLMs**: FeatEng is the first benchmark specifically designed to evaluate LLMs' capabilities in feature engineering, a complex task that integrates domain knowledge, reasoning, and coding skills.
2. **Practical Relevance**: By focusing on feature engineering, FeatEng assesses LLMs' practical utility in real-world data science workflows, potentially paving the way for LLMs to assist human data scientists.
3. **Correlation with Human Judgment**: The benchmark demonstrates a strong correlation with Chatbot Arena ELO scores, suggesting that it captures essential aspects of LLM performance relevant to human users, even though it was not explicitly designed for this purpose.
4. **Dataset Diversity**: The use of 103 diverse datasets from various domains ensures the benchmark's broad applicability and reduces the risk of overfitting.

#### Weaknesses
1. **Narrow Scope (nK3S)**: The focus on feature engineering might be perceived as a narrow application, potentially limiting the generalizability of the findings to broader LLM capabilities.
2. **Overfitting and Data Contamination Concerns (Eugm, YFRZ)**: The use of datasets from Kaggle raises concerns about potential overfitting or data contamination if models have been trained on similar data.
3. **Dependence on XGBoost (Eugm, KB4t)**: The sole use of XGBoost for evaluation might limit the benchmark's generality, as different models might respond differently to feature engineering efforts.
4. **Lack of Human Baselines (Eugm, KB4t)**: The absence of human performance benchmarks makes it challenging to contextualize the results and understand the practical significance of LLM performance.
5. **Single-Pass Evaluation (KB4t)**: The single-pass evaluation method contrasts with iterative real-world data science practices, potentially underestimating LLM capabilities that could improve with iterative refinement.
6. **Limited Literature Review and Comparisons (YFRZ)**: The paper's discussion of existing benchmarks is seen as vague and lacks specific experimental evidence or comparisons to concurrent works evaluating LLMs in complex scenarios.

**Additional Comments On Reviewer Discussion:**

1. **Scope and Significance of Feature Engineering (nK3S):**
   - **Concern**: Feature engineering is a narrow application, not reflecting general intelligence.
   - **Response**: The authors argue that feature engineering requires a synthesis of domain knowledge, reasoning, creativity, and technical skills, making it a microcosm of complex problem-solving. They expanded the experimental section (Section 3) with qualitative and quantitative analyses, demonstrating the task's depth and relevance. Examples of innovative feature engineering by LLMs were provided to highlight the task's complexity.

2. **Clarity and Presentation Improvements (KB4t, YFRZ):**
   - **Concern**: Presentation of results and figures needed clarity and additional visualizations.
   - **Response**: The manuscript was updated with enhanced tables (e.g., Table 1 with color-coded metrics) and new figures (e.g., line plots in Appendix Figures 7 and 8). Figure 3 was replaced with a clearer flowchart (Figure 4) to illustrate the evaluation pipeline. These changes improved the paper's readability and interpretability.

3. **Result Interpretation and Visualization (KB4t):**
   - **Concern**: Limited interpretation of results, with suggestions for additional baselines and parameter scaling analyses.
   - **Response**: The authors included more detailed analyses in the revised manuscript, covering three pages (pages 7-10). They provided insights into feature types and their impact on performance, showing that larger models exhibit variability in improvements. While additional AutoML baselines were not included, their potential value was acknowledged for future work.

4. **Potential Overfitting and Data Contamination (Eugm, YFRZ):**
   - **Concern**: Models might overfit or have been exposed to similar data during training.
   - **Response**: The authors mitigated these concerns by using diverse datasets (103 from various domains) and modifying them to reduce prior exposure likelihood. They emphasized the open-ended nature of feature engineering, which demands creativity rather than memorization. These points were clarified in Section 2.1 and Table 5.

5. **Dependence on XGBoost as the Sole Evaluation Model (Eugm, KB4t):**
   - **Concern**: Relying on XGBoost might limit the benchmark's generality.
   - **Response**: XGBoost was chosen for its robustness and efficiency, ensuring consistent evaluations across datasets. The authors acknowledged the potential benefit of including other models and noted this as future work.

6. **Inclusion of Human Baselines (Eugm, KB4t):**
   - **Concern**: Lack of human baselines makes it difficult to contextualize model performance.
   - **Response**: The authors recognized the importance of human baselines and proposed a community-driven approach to collect human-generated solutions over time, as mentioned in Section 6.

7. **Literature Review and Comparison to Existing Benchmarks (YFRZ):**
   - **Concern**: Vague discussion of existing benchmarks and lack of comparisons to concurrent works.
   - **Response**: The authors expanded Section 1.1 and Appendix H with concrete examples and discussions of concurrent works, emphasizing FeatEng's unique focus on integrated skill assessment. They also conducted a correlation analysis with Chatbot Arena ELO scores, demonstrating FeatEng's alignment with human judgment and its cost-effectiveness compared to other benchmarks.

8. **Single- vs Multi-pass Evaluation (KB4t):**
   - **Concern**: Single-pass evaluation might not reflect real-world iterative practices.
   - **Response**: The authors clarified that the single-pass method was chosen for simplicity and fairness, while acknowledging that iterative approaches could enhance performance. They encouraged researchers to use FeatEng in more interactive contexts, noting the strong correlation with Chatbot Arena ELO as validation of the single-pass method's reasonableness.

#### Final Decision Weighting
The authors demonstrated a strong commitment to addressing reviewer feedback, significantly improving the manuscript through detailed analyses, clarifications, and visual enhancements. While some concerns about the benchmark's scope and potential overfitting remain, the authors provided reasonable justifications and mitigation strategies. The addition of correlation analyses with Chatbot Arena ELO scores and the proposal for community involvement in collecting human baselines strengthen the paper's contributions.

---

### Decision · Program_Chairs · 2025-01-22

Reject